# Analytical Performance of a Highly Sensitive System to Detect Gene Variants Using Next-Generation Sequencing for Lung Cancer Companion Diagnostics

**DOI:** 10.3390/diagnostics13081476

**Published:** 2023-04-19

**Authors:** Kikuya Kato, Jiro Okami, Harumi Nakamura, Keiichiro Honma, Yoshiharu Sato, Seiji Nakamura, Yoji Kukita, Shin-ichi Nakatsuka, Masahiko Higashiyama

**Affiliations:** 1Laboratory of Medical Genomics, Nara Institute of Science and Technology, Nara 630-0192, Japan; 2Department of General Thoracic Surgery, Osaka International Cancer Institute, Osaka 540-0008, Japan; 3Laboratory of Genomic Pathology, Osaka International Cancer Institute, Osaka 540-0008, Japan; 4Department of Diagnostic Pathology and Cytology, Osaka International Cancer Institute, Osaka 540-0008, Japan; 5DNA Chip Research Inc., Tokyo 105-0022, Japan; 6Department of Thoracic Surgery, Higashiosaka City Medical Center, Osaka 578-8588, Japan

**Keywords:** non-small cell lung carcinoma, molecular targeted therapy, companion diagnostics, next-generation sequencing, gene panel

## Abstract

The recent increase in the number of molecular targeted agents for lung cancer has led to the demand for the simultaneous testing of multiple genes. Although gene panels using next-generation sequencing (NGS) are ideal, conventional panels require a high tumor content, and biopsy samples often do not meet this requirement. We developed a new NGS panel, called compact panel, characterized by high sensitivity, with detection limits for mutations of 0.14%, 0.20%, 0.48%, 0.24%, and 0.20% for *EGFR* exon 19 deletion, L858R, T790M, *BRAF* V600E, and *KRAS* G12C, respectively. Mutation detection also had a high quantitative ability, with correlation coefficients ranging from 0.966 to 0.992. The threshold for fusion detection was 1%. The panel exhibited good concordance with the approved tests. The identity rates were as follows: *EGFR* positive, 100% (95% confidence interval, 95.5–100); *EGFR* negative, 90.9 (82.2–96.3); *BRAF* positive, 100 (59.0–100); *BRAF* negative, 100 (94.9–100); *KRAS* G12C positive, 100 (92.7–100); *KRAS* G12C negative, 100 (93.0–100); *ALK* positive, 96.7 (83.8–99.9); *ALK* negative, 98.4 (97.2–99.2); *ROS1* positive, 100 (66.4–100); *ROS1* negative, 99.0 (94.6–100); *MET* positive, 98.0 (89.0–99.9); *MET* negative 100 (92.8–100); *RET* positive, 93.8 (69.8–100); *RET* negative, 100 (94.9–100). The analytical performance showed that the panel could handle various types of biopsy samples obtained by routine clinical practice without requiring strict pathological monitoring, as in the case of conventional NGS panels.

## 1. Introduction

One of the major paradigms in modern cancer therapy involves predicting the efficacy of anticancer agents based on genomic information of a patient. The most notable example of this approach is therapy for advanced non-small cell lung carcinoma (NSCLC). Companion diagnostics are used to select molecular targeted agents for patients based on mutations in the target genes. The era of precision medicine began with the discovery of a correlation between *epidermal growth factor receptor* (*EGFR*) mutations and EGFR-targeted treatment efficacy [1]. Since then, the number of molecular targeted agents for treating NSCLC has continued to grow. The current molecular targeted agents for companion diagnostics, including osimertinib [2], alectinib [3], crizotinib [4], dabrafenib plus trametinib [5], tepotinib [6], selpercatinib [7], and sotorasib [8], target EGFR, ALK, ROS1, BRAF, MET, RET, and KRAS, respectively. The latest American Society of Clinical Oncology (ASCO) living guidelines for stage IV NSCLC with driver alterations [9] include these genes and NTRK. Furthermore, other agents targeting EGFR with exon 20 insertions [10] will be available soon.

Previously, only EGFR tyrosine kinase inhibitors were available, and diagnostic tests for a single gene were sufficient for practical use. However, with the growing number of agents, there is a strong demand for the simultaneous testing of multiple genes. In this context, next-generation sequencing (NGS) panels, which are analytical/diagnostic systems that detect variants in multiple cancer-related genes using NGS, such as Foundation One CDx (Foundation Medicine, Inc., Cambridge, MA, USA) and Oncomine Dx Target Test (Thermo Fisher Scientific, Inc., Waltham, MA, USA), should be helpful.

However, these panels may not be optimally designed for diagnostic purposes. The panel prototypes mentioned above were intended for biomarker exploration, with a priority placed on the coverage of cancer-related genes rather than on the sensitivity and accuracy of variant detection. The primary limitation of these diagnostic applications is the requirement for a high tumor content (>20%) of the samples for diagnosis. As a result, many samples cannot be tested, and single-gene tests based on real-time polymerase chain reaction (PCR) must be used. While these NGS panels work well with samples collected under ideal conditions, such as in clinical trials, they are not optimized for testing the samples obtained in routine clinical practice. This shortcoming is mainly due to the low sensitivity of variant detection.

To address this issue, we aimed to develop an NGS panel, named compact panel, for companion diagnosis of NSCLC. The compact panel achieved a higher sensitivity than conventional NGS panels and single-gene tests. This panel was approved as a medical device by the Ministry of Health, Labour, and Welfare of Japan on 16 November 2022.

## 2. Materials and Methods

### 2.1. DNA and RNA Samples

*Sensitivity tests.* DNA samples, including various fractions of mutant alleles, were prepared using solid tumor analysis reference standards (Horizon Discovery Ltd., Lafayette, CO, USA): HDx FFPE *EGFR* e19 del (50%), HDx FFPE *EGFR* L858R (50%), FFPE *EGFR* T790M (50%), and HDx FFPE *BRAF* V600E (50%). Wild-type formalin-fixed paraffin-embedded (FFPE) genomic DNA from the lungs, including the bronchioles (Cureline Inc., Brisbane, CA, USA), was used to adjust for allele frequencies. For *MET* exon 14 skipping, synthesized DNA was used, skipping the need for reverse transcription. RNA samples, including fusions, were constructed using the *ALK–RET–ROS1*-targeted FFPE RNA Fusion Reference Standards (Horizon Discovery Ltd., Lafayette, CO, USA). Wild-type FFPE RNA from the lungs, including the bronchioles (Cureline Inc., Brisbane, CA, USA), was used to adjust the RNA concentration.

*Concordance tests.* FFPE NSCLC and frozen unfixed samples were obtained through surgical resection and stored at the Osaka International Cancer Institute. DNA and RNA were then purified from these samples using a Maxwell RSC instrument (Promega Corporation, Madison, WI, USA).

### 2.2. Library Preparation and Sequencing of DNA Modules

PCR amplification was performed in 50 µL of reaction mixture containing 1 × buffer (Toyobo, Inc., Osaka, Japan), 0.2 mM dNTPs, 1.5 mM Mg_2_SO_4_, 5 ng of DNA purified from FFPE NSCLC, 0.3 µM mixtures of each primer, and 0.02 U of KOD-Plus-Neo (Toyobo, Inc., Osaka, Japan). The cycle conditions were 98 °C for 10 s and 62 °C (module I) or 68 °C (module II) for 30 s for forty cycles. The amplified products were then purified using AMPure XP (Beckman Coulter Life Sciences, Loveland, CO, USA) and subjected to library construction using the MiSeq System (Illumina, Inc., San Diego, CA, USA) and GenNext NGS Library Prep Kit (Toyobo, Inc., Osaka, Japan). Index sequences were introduced to differentiate between individual samples using the TruSeq DNA Single Indexes Set A or TruSeq DNA CD Indexes Set (Illumina, Inc., San Diego, CA, USA). Sequencing was performed using the MiSeq System (Illumina, Inc., San Diego, CA, USA), with a minimum of 5000 reads per fragment.

### 2.3. Library Preparation and Sequencing of RNA Modules

Reverse transcription was performed in 20 µL of reaction mixture containing 1 × ReverTra buffer (Toyobo, Inc., Osaka, Japan), 1 mM dNTPs, 10 ng of RNA purified from FFPE NSCLC, and 100 U of ReverTra Ace (Toyobo, Inc., Osaka, Japan). After denaturation with RNA and a 9-base random primer (Toyobo, Inc., Osaka, Japan) at 65 °C for 5 min, the reaction mixture was incubated at 30 °C for 10 min and then at 42 °C for 60 min. PCR amplification was then performed in 50 µL of reaction mixture containing 1 × buffer (Toyobo, Inc., Osaka, Japan), 0.4 mM dNTPs, 1.5 mM Mg_2_SO_4_, the 20 µL reaction mixture, 0.25 µM mixtures of each primer, and 1 U of KOD-Fx-Neo (Toyobo, Inc., Osaka, Japan). The cycle conditions were 98 °C for 15 s, 60 °C for 30 s, and 68 °C for 10 s for forty cycles, followed by extension at 68 °C for 1 min. After purification with AMPure XP (Beckman Coulter Life Sciences, Loveland, CO, USA), the amplified products were subjected to library construction and sequencing as described in the previous section. A minimum of 300 reads per sample was required.

### 2.4. Sequence Data Analysis

Adapter sequences were removed from the output sequences of the MiSeq System using Trimmomatic v. 0.33 [11]. Then, paired-end reads were merged using FLASH v. 1.2.11 [12]. Finally, the resulting sequences were aligned to the reference sequences using BWA v. 0.7.17 [13].

### 2.5. Analysis of Discordant Samples

Digital PCR was performed using the QX200 Droplet Digital PCR System (Bio-Rad Laboratories Inc., Hercules, CA, USA), following the supplier’s protocol. The non-overlapping integrated read sequencing system (NOIR-SS) is a high-fidelity target sequencing system that uses molecular barcodes with the ability to quantify mutations [14]. We performed the NOIR-SS assay as described previously.

### 2.6. Preparation of Artificial Samples

*BRAF.* The artificial *BRAF* mutation-positive sample consisted of DNA purified from *BRAF* V600E-positive and mutation-negative NSCLC samples. Artificial samples with various mutant allele fractions were generated by changing the DNA ratio from mutation-positive to mutation-negative samples. The mutant allele fractions of the five *BRAF* V600E-positive NSCLC samples were 16.2%, 14.2%, 14.1%, 16.3%, and 14.2%. For each mutation-positive artificial sample, we used one mutation-negative NSCLC sample. We generated 80 artificial samples using 60 FFPE and 20 unfixed NSCLC mutation-negative samples. The mutant allele fraction of the artificial samples ranged from 16.2% to 10%. The lower limit of the fraction was determined based on the detection limit of the Oncomine Dx Target Test.

*ROS1.* The artificial *ROS1* fusion-positive sample was generated using RNA purified from *ROS1* fusion-positive and fusion-negative NSCLC samples. The fusion allele fractions were varied by changing the ratio of RNA of the fusion-positive and fusion-negative samples. The variants were identified as SD2_R32 (three FFPE and one unfixed sample), CD6_R34 (three FFPE samples), SD4_R32 (one FFPE sample), EZ10_R34 (one FFPE sample), and SL4_R32 (Horizon standard RNA). Artificial samples were generated from each fusion-positive FFPE NSCLC sample with fusion-positive RNA fractions of 9%, 12%, 16%, 20%, 24%, 28%, 32%, 36%, 40%, and 90%, by mixing with fusion-negative FFPE NSCLC samples. Similarly, from the unfixed fusion-positive NSCLC sample and the Horizon RNA standard, artificial samples with fusion-positive RNA fractions of 2%, 4%, 6%, 8%, 10%, 12%, 14%, 16%, 18%, and 90% were generated by mixing with fusion-negative unfixed NSCLC samples. Using 80 FFPE fusion-negative NSCLC samples and 20 unfixed fusion-negative NSCLC samples, 100 artificial samples were generated.

*RET.* An artificial *RET* fusion-positive sample was prepared using RNA purified from *RET* fusion-positive and fusion-negative NSCLC samples. Artificial samples with various fusion allele fractions were generated by changing the ratio of RNA of the fusion-positive and fusion-negative samples. The 17 fusion-positive NSCLC samples had the following variants: *KIF5B* exon 15–*RET* exon 12 (*n* = 11), *KIF5B* exon 22–*RET* exon 12 (*n* = 2), *KIF5B* exon 24–*RET* exon 11 (*n* = 1), *CCDC6* exon 1–*RET* exon 12 (*n* = 3). All samples except for two *KIF5B* exon 15–*RET* exon 12 frozen unfixed tissues were FFPE. Four artificial samples with fusion-positive RNA fractions of 85%, 90%, 95%, and 100% were generated from each fusion-positive FFPE NSCLC sample. In addition, four samples with 80%, four with 85%, and two with 90% fusion-positive RNA fraction were generated from each fusion-positive unfixed NSCLC sample. In total, 80 artificial samples were generated.

Concordance tests with artificial samples and conventional concordance test were performed simultaneously.

## 3. Results

### 3.1. Design of the Compact Panel

The compact panel was designed to reduce the number of target genes and improve the panel design flexibility for genetic diagnosis for patients with NSCLC by including target genes of agents that have been already approved or are expected to be approved soon. DNA was used for detecting mutations, while RNA was used for detecting fusion genes. This is because when DNA is used for detecting fusion genes, it is necessary to determine the sequence of the intron region extensively, whereas in the case of RNA, only the sequence of the fusion site is required. To evaluate the analytical performance of each template preparation step in the workflow of variant detection using NGS, we introduced the concept of “module”, where a module refers to a template preparation step performed using a single PCR reaction mixture. For instance, DNA module I corresponds to a PCR reaction mixture that amplifies fragments containing the *EGFR* major mutations, *BRAF* V600E, and *KRAS* G12C. The evaluation of DNA module I revealed the analytical performance for the detection of these variants. The panel consisted of four modules: two DNA modules and two RNA modules (Table 1), using DNA and RNA as templates, targeting specific variants as follows: DNA module I for *EGFR* exon 19 deletion, *EGFR* L858R, *BRAF* V600E, and *KRAS* G12C; DNA module II for other *EGFR* mutations, *HER2* exon 20, and *MET* exon 14 skipping; RNA module I for *ALK* fusion and *MET* exon 14 skipping; and RNA module II for *ROS1* and *RET* fusion genes. As uncommon single or compound *EGFR* mutations may have therapeutic importance [15], we covered the major parts of exons 18–21 with DNA modules I and II. The covered areas were as follows: exon 18, N700–G721 (DNA module II); exon 19, I744-I759 (DNA module I); exon 20, T785-L799 (DNA module I), E762–C781 (DNA module II); exon 21, H850–G863 (DNA module I), D830–P848 (DNA module II). Detailed lists of *ALK*, *ROS1*, and *RET* fusion variants are provided in Appendix A. The modular structure of the panel simplifies obtaining official authorization for revising the panel; an analytical performance evaluation is only required for a new module without the need to re-evaluate the entire panel, because the biochemical reactions in a module do not interfere with those in other modules. The replacement or addition of modules can help identify new diagnostic genes in the future.

To increase the sensitivity, we implemented different strategies for mutation and fusion detection. To detect mutations, templates were amplified from genomic DNA, and deep sequencing was performed, that is, repeated sequencing of target regions. The sensitivity of variant detection depends on the number of sequence reads [16], and thus, we increased the number to 5000 reads instead of 700 reads as used in the Oncomine Dx Target Test. To detect fusions, we used multiplex PCR to amplify the junctions of known variants using RNAs as templates. As there is a considerable number of counterpart genes, a single reaction mixture is required to include multiple primers, often leading to low sensitivity. Therefore, we reduced the number of primers in a single reaction mixture to enhance the sensitivity. To determine the optimal combination of PCR primers, we tested multiple combinations. The final combination used in the RNA modules I and II achieved high sensitivity.

PCR primers were designed so that the sizes of the amplified products were <110 base pairs except three amplicons, which allowed PCR amplification using deteriorated DNA/RNA templates. The primer sequences are listed in Appendix A. We used derivatives of the KOD DNA polymerase, an archaeal family B DNA polymerase, for PCR amplification. The 3′–5′ proofreading exonuclease of archaeal family B DNA polymerase hinders the copying of template strand deaminated bases [17]; therefore, false-positive mutations generated during formalin fixation are expected to be excluded. A flowchart depicting the analytical procedure is shown in Figure 1.

### 3.2. Sensitivity of Mutation Detection Using DNA as a Template

The thresholds for mutation detection were set using anomaly detection [16]. To perform anomaly detection, the probability of false positives was estimated from the measured values of 56 normal samples, assuming a Poisson distribution. Threshold values were set so that the probability of false positives was 10^−10^. These values were defined by allele frequency (Table 2).

To evaluate the sensitivity of DNA module I, we analyzed 24 samples with 1% and 0% mutant allele DNA in each test. There were no false positives, i.e., no mutation positives in 0% mutated DNA, and no false negatives, i.e., no mutation negatives in 1% mutated DNA (Table 2).

### 3.3. Sensitivity of Fusion Detection Using RNA as a Template

The thresholds for fusion detection were established using anomaly detection. The probability of false positives was estimated from the measured values of 48 normal samples. The threshold values were set to ensure a 10^−5^ probability of false positives based on the Poisson distribution. The values were defined by the tumor mutation (TM) score, which represents the number of positive reads per 100,000 reads, and are provided in Table 2.

We prepared fusion-positive samples containing 1% RNA derived from fusion-positive cells to assess the panel’s sensitivity for fusion detection. We tested 24 or 48 samples with fusion-positive and fusion-negative RNA for each sensitivity test. The false positive and negative rates were <0.5% (Table 2).

### 3.4. Quantification of Mutation Detection

The quantitative ability of the DNA module I was examined using artificial DNA samples prepared such that 1−8% of the total DNA contained mutant alleles. The results are shown in Figure 2.

The data showed a strong linear relationship between the numbers of mutant alleles inoculated and the frequency of mutant alleles observed using deep sequencing. The correlation coefficients for *EGFR* exon 19 deletion, *EGFR* L858R, *EGFR* T790M, *BRAF* V600E, and *KRAS* G12C were 0973, 0.972, 0.966, 0.992, and 0.991, respectively.

### 3.5. Concordance with Conventional Diagnostic Tests

The performance of the compact panel was compared to that of the approved diagnostic tests. The reference diagnostic tests used were as follows: Cobas EGFR Mutation Test v2 (Roche Diagnostics K.K., Tokyo, Japan) for *EGFR*, Histofine ALK iAEP kit (Nichirei Bioscience, Inc., Tokyo, Japan), and Vysis ALK Break Apart FISH Probe Kit (Abbott Laboratories, Chicago, IL, USA) for *ALK*, ArcherMET (Archer DX, Inc., Boulder, CO, USA) for *MET*, therascreen KRAS RGQ PCR Kit (Qiagen N.V., Venlo, The Netherlands) for *KRAS* G12C, and Oncomine Dx Target Test for *BRAF* and *RET*. FFPE NSCLC samples were analyzed simultaneously using the compact panel and the reference test, and the concordance of the two tests was examined. The results are shown in Table 3, and good concordance between the two tests was observed, demonstrating the practical feasibility of the compact panel.

Seven samples showed discordance between the compact panel and the reference test for *EGFR* mutation analysis, with the compact panel testing positive and the reference test being negative. However, when analyzed using digital PCR, all seven samples were found to be mutation-positive, consistent with the results of the compact panel. One *ALK* discordant sample was analyzed using the non-overlapping integrated read sequencing system (NOIR-SS) [14] and it was found to be an *ALK* fusion with *C2ORF71*, a rare fusion type not covered in either the compact panel or the Oncomine Dx Target Test. For the other discordant samples, the cause of the discrepancy is not clear.

The wide range of 95% confidence intervals for *BRAF* V600E, *ROS1* fusion, and *RET* fusion suggests ambiguity in estimating their positive identity rates due to the small number of clinical samples. The incidence of these variants is low, making it challenging to collect NSCLC samples with such variants, even from a large cancer tissue repository. Therefore, we introduced a new concordance test using artificial samples to address this. These samples were mixtures of nucleic acids purified from NSCLC tissues with and without variants, providing both quantitative and qualitative variations. The variant allele fractions were simulated by changing the ratio of variant-positive and variant-negative samples. In addition, different variant-negative samples were used to retain various qualities. The results are presented in Table 4 and showed excellent concordance for all three variants.

### 3.6. Incidence of Mutations and Fusions

The NSCLC samples stored at the Osaka International Cancer Institute were screened using the compact panel, and the incidence of mutations and fusions at the population level was determined (Table 5). The results were compared with those obtained from the Oncomine Comprehensive Assay in a nationwide project named Scrum-Japan [18]. The *EGFR* mutation-positive samples were excluded from the Osaka International Cancer Institute cohort because Scrum-Japan collected only *EGFR* mutation-negative samples. The observed incidence was higher for *KRAS*, *BRAF*, *ALK*, and *MET* in the Osaka International Cancer Institute cohort, while for *ROS1*, it was higher in the Scrum-Japan cohort.

## 4. Discussion

We developed a compact panel that allows the simultaneous testing of multiple genes for companion diagnostics of NSCLC using NGS. The compact panel was designed to include target genes of agents already approved or expected to be approved soon, providing flexibility in panel design for genetic diagnosis. In addition, we introduced the concept of “module” to simplify the process of obtaining official authorization for panel revision and to identify new diagnostic genes in the future.

The presence or absence of driver mutations is crucial for therapeutic decision making in treating advanced NSCLC. For example, the ASCO living guidelines were issued for NSCLC with [9] and without [19] driver alterations. The guidelines for NSCLC with driver alterations focus on molecular targeted reagents. In contrast, those for NSCLC without driver alterations focus on immune checkpoint inhibitors. Thus, the diagnostic tool used to examine driver alterations plays a definitive role in therapeutic decision making for NSCLC. However, the current tools have problems when used in routine clinical practice.

Companion diagnostics for advanced NSCLC have historically been used for individual genes, such as *EGFR* mutations using real-time PCR-based tests and *ALK* fusions using in situ hybridization. However, as the number of target genes increases, there is a growing demand for the simultaneous testing of multiple genes. NGS is an ideal tool for the simultaneous analysis of multiple genes. In the USA, NGS panels have been used for 44% of patients with NSCLC who required genetic testing [20].

However, the introduction of NGS panels in Japanese medical practice poses unique challenges. *EGFR* mutations are estimated to be 12.8% in Europeans but 49.1% in Asians [21]; the high prevalence of *EGFR* mutations is one reason for the emphasis on genetic testing in Japan. Another reason is the difference in specimen collection methods. NSCLC samples for genetic testing in the USA are usually obtained by core needle biopsy or transthoracic fine-needle aspiration biopsy guided by computed tomography or ultrasound [22]. In contrast, biopsies using bronchoscopy are more prevalent in Japan as physicians avoid invasive procedures. This difference in biopsy practices is another obstacle to introducing NGS panels. In Japan, the feasibility of NGS panels is generally lower than that of other diagnostics in clinical practice, especially regarding nonsurgical biopsy [23]. In addition, the success rate of the Oncomine Dx Target Test is influenced by tissue size and tumor cell count [24,25]. Consequently, strict pathological monitoring is necessary to select the samples for testing, leading to the exclusion of a significant fraction of samples. Furthermore, the requirement for specialized skills in pathology to perform the monitoring may lead to expanding medical disparities among facilities.

The development of a high-sensitivity NGS panel can address the challenges faced in routine clinical practice by simultaneously testing multiple genes with a lower requirement for strict pathological monitoring. Our compact panel has a sensitivity that allows detecting mutation/fusion in samples with 1% tumor content. The confirmed detection limit of DNA module I is lower than that of the Oncomine Dx Target Test (6–13%) and Cobas EGFR Mutation Test v2 (1.26–6.81%) (FDA Summary of Safety and Effectiveness Data). With only 5 ng of DNA/RNA necessary for the test, a smaller amount of tissue is needed from a routine biopsy. Although initially designed for the Japanese medical environment, the high-sensitivity NGS panel can also benefit other medical environments. Studies using cytological samples have demonstrated the stable performance of the compact panel in a real clinical setting [26,27,28], which was not possible with conventional NGS panels.

Liquid biopsy using plasma DNA is a promising noninvasive diagnostic procedure. However, it has a major limitation as a companion diagnostic method. A significant proportion of patients with advanced NSCLC do not release circulating tumor DNA (ctDNA), and there are no methods to discriminate between patients with and without ctDNA. In our previous study, approximately 30% of patients with advanced NSCLC did not have ctDNA [29]. Tissue biopsy has a clear advantage over liquid biopsy, as tumor cells can be confirmed through pathological or cytological examination. Assuming that NSCLC samples subjected to the compact panel undergo pathological/cytological examination, we have set the panel’s sensitivity to 1%. Maintaining an excessive sensitivity can cause stress on the workflow, although the detection limit of variant detection with NGS can be as low as 0.001% [14].

Recently, a multicenter, international, phase 2 study demonstrated the effectiveness of trastuzumab deruxtecan in patients with metastatic *HER2*-mutant NSCLC refractory to standard treatment [30]. *HER2* is already included in DNA module II, and its variant detection may be applied to companion diagnostics. A compact panel may not include a new target gene in the future. Large NGS panels such as Foundation One CDx will likely contain such genes. However, diagnostic applications’ feasibility depends on a particular gene’s analytical performance. The compact panel can add a new module with suitable analytical performance. Thus, the small coverage of genes is not disadvantageous for diagnostic applications.

FFPE is essential for pathological diagnosis. FFPE tissues have been the standard sample type for genetic diagnosis. However, formalin fixation can cause DNA damage due to the formation of cross-links between DNA and proteins, single-strand breaks in DNA, and inhibition of the resealing of single-strand breaks produced by ionizing radiation [31]. To enhance the accuracy of a genetic diagnosis, it is desirable to avoid formalin fixation.

To prevent the degradation of nucleic acids, unfixed tissues must be frozen, which is often difficult in some facilities. The ammonium sulfate solution allows the efficient preservation of RNA/DNA for several days without refrigeration [32] and is expected to act as a preservative for unfixed tissues. This would be particularly useful for washing samples obtained via cytodiagnostic brushing or curette washing.

In conclusion, we have developed an NGS panel called compact panel with high sensitivity and accommodative to various NSCLC samples without requiring strict pathological monitoring. The compact panel can potentially improve the diagnosis and treatment of NSCLC and benefit medical environments beyond Japan. Further studies are needed to validate the panel’s performance in larger cohorts and investigate its utility in clinical practice.

## Figures and Tables

**Figure 1 diagnostics-13-01476-f001:**
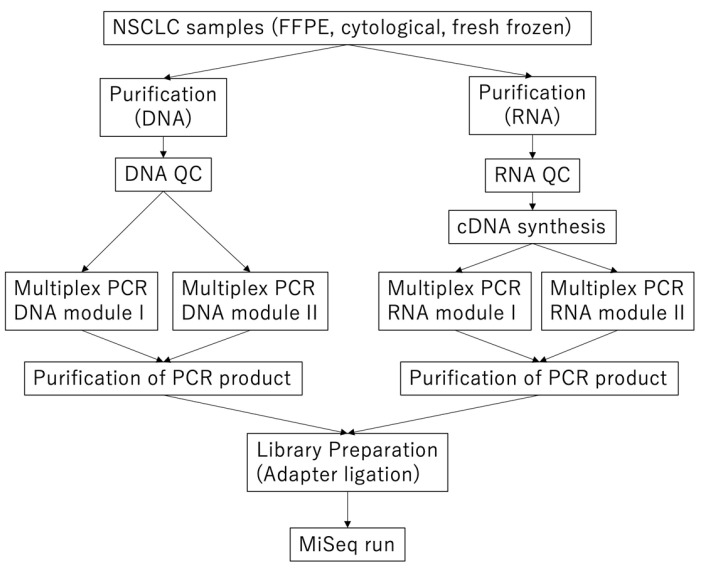
Flowchart depicting the analytical procedure.

**Figure 2 diagnostics-13-01476-f002:**
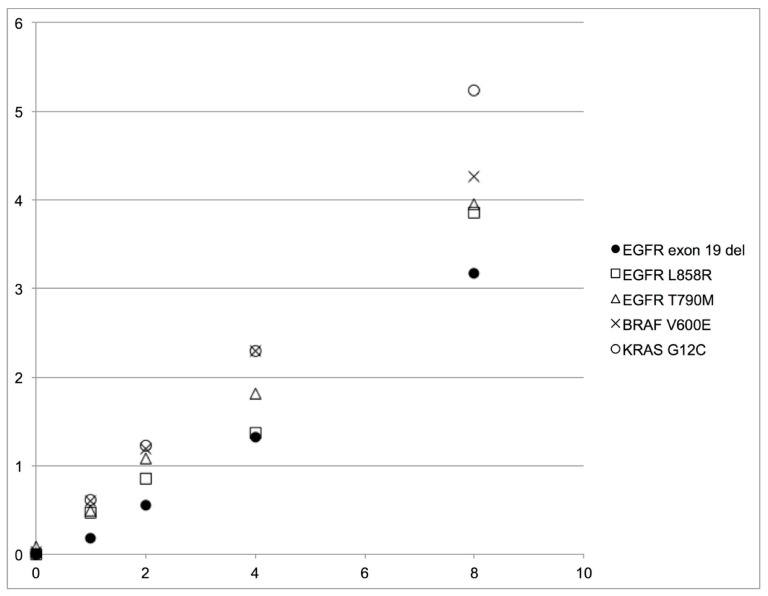
Quantitative ability of DNA module I. Horizontal axis, mutation allele frequency of artificial DNA (%); vertical axis, mutation allele frequency measured with the compact panel (%). Each data point is an average of eight samples.

**Table 1 diagnostics-13-01476-t001:** Modules of the compact panel.

	Gene	Target Mutations/Fusion Variants
DNA module I	*EGFR*	Exon 19 deletion, L858R, T790M, L861Q, L861R
	*BRAF*	V600E
	*KRAS*	G12C
DNA module II	*EGFR*	G719X, E709X, S768I, exon 20 insertion
	*HER2*	Exon 20 mutations
	*MET*	Exon 14 skipping
RNA module I	*ALK*	*EML4*, 22 variants; *KIF5*, 3; *TFG*, 1; *HIP*, 3; *KLC1*, 1
	*MET*	Exon 14 skipping
RNA module II	*ROS1*	*CD74*, 5 variants; *SLC34A2*, 7; *EZR*, 1; *GOPC*, 2; *SDC4*, 4; *LRIG*, 1; *TPM3*, 1; *CCDC6*, 1; *KDELR2*, 1
	*RET*	*KIF5B*, 7 variants; *CCDC6*, 1; *NCOA4*, 1

**Table 2 diagnostics-13-01476-t002:** Sensitivity test of the compact panel. The number of samples analyzed was 24, except for the *ALK* false negative (48 samples).

	Threshold of Detection DNA, % Allele Frequency; RNA, TM Score	Sensitivity Test	
		False Negative	False Positive
DNA module			
*EGFR* exon 19 del	0.14	0	0
*EGFR* L858R	0.2	0	0
*EGFR* T790M	0.48	0	0
*BRAF* V600E	0.24	0	0
*KRAS* G12C	0.2	0	0
RNA module			
*ALK* fusion	188	0	1
*ROS1* fusion	32	0	1
*RET* fusion	18	0	0
*MET* exon 14 skipping	28	0	0

**Table 3 diagnostics-13-01476-t003:** Concordance of the compact panel with approved diagnostic tests. Parentheses in identity rates indicate a 95% confidence interval.

		Compact Panel Positive	Compact Panel Negative	Positive Identity Rate	Negative Identity Rate	Reference Test
*EGFR*	Reference test positive	73	0	100 (95.5–100)	-	Cobas EGFR Mutation Test v2
	Reference test negative	7	70	-	90.9 (82.2–96.3)	
*BRAF* V600E	Reference test positive	7	0	100 (59.0–100)	-	Oncomine Dx Target Test
	Reference test negative	0	70	-	100 (94.9–100)	
*KRAS* G12C	Reference test positive	49	0	100 (92.7–100)	-	therascreen KRAS RGQ PCR Kit
	Reference test negative	0	51	-	100 (93.0–100)	
*ALK*	Reference test positive	29	1	96.7 (83.8–99.9)	-	Histofine ALK iAEP kit/Vysis ALK Break Apart FISH Probe Kit
	Reference test negative	11	692	-	98.4 (97.2–99.2)	
*ROS1*	Reference test positive	9	0	100 (66.4–100)	-	OncoGuide AmoyDx ROS1
	Reference test negative	1	99	-	99.0 (94.6–100)	
*MET*	Reference test positive	48	1	98.0 (89.0–99.9)	-	ArcherMET
	Reference test negative	0	50	-	100 (92.8–100)	
*RET*	Reference test positive	15	1	93.8 (69.8–100)	-	Oncomine Dx Target Test
	Reference test negative	0	70	-	100 (94.9–100)	

**Table 4 diagnostics-13-01476-t004:** Concordance of the compact panel with approved diagnostic tests using artificial samples. Parentheses in identity rates indicate a 95% confidence interval.

		Compact Panel Positive	Compact Panel Negative	Positive Identity Rate	Reference Test
*BRAF* V600E	Reference test positive	77	0	100 (95.3–100)	Oncomine Dx Target Test
	Reference test no call	3	0	-	
*ROS1*	Reference test positive	99	1	99.0 (94.6–100)	OncoGuide AmoyDx ROS1
*RET*	Reference test positive	79	1	98.8 (93.2–100)	Oncomine Dx Target Test

**Table 5 diagnostics-13-01476-t005:** Frequencies of mutations and fusions among *EGFR* mutation-negative samples. Parentheses in percent frequency indicate the 95% confidence interval. Samples from the Osaka International Cancer Institute were obtained from 3 February 2011 to 24 August 2020. The number of *BRAF* V600E and *KRAS* G12C samples was 11 and 77, respectively.

	Osaka International Cancer Institute		Scrum-Japan	
Assay system	Compact panel		Oncomine comprehensive assay	
Total number of samples	827		3919	
Gene	Number of variant-positive samples	Percent frequency	Number of variant-positive samples	Percent frequency
*KRAS*	202	24.4 (21.5–27.5)	382	9.7 (8.8–10.7)
*BRAF*	36	4.4 (3.1–6.0)	97	2.5 (2.0–3.0)
*ALK*	41	5 (3.6–6.7)	97	2.5 (2.0–3.0)
*ROS1*	10	1.2 (0.6–2.2)	142	3.6 (3.1–4.3)
*MET*	56	6.8 (5.2–8.7)	98	2.5 (2.0–3.0)
*RET*	18	2.2 (1.3–3.4)	100	2.6 (2.1–3.1)

## Data Availability

Data are contained within the article and Appendix A.

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
