# Peer review of "Analytical Performance of a Highly Sensitive System to Detect Gene Variants Using Next-Generation Sequencing for Lung Cancer Companion Diagnostics"

_diagnostics, 2023, doi:10.3390/diagnostics13081476_

Round 1

Reviewer 1 Report

This is an excellent article describing a diagnostic panel designed specifically for clinical practices. Approval of the test by the Japanese government is a testament to its effectiveness. I have a few minor suggestions:

1.       Can the authors include a brief background explaining why some mutations are detected through DNA and others through RNA?

2.       This compact panel can have global impact beyond Japan. Are the variants tested in this panel occurring at similar frequencies in other countries? Such data can demonstrate how applicable the test is outside Japan.

Author Response

  1. Can the authors include a brief background explaining why some mutations are detected through DNA and others through RNA?

Our response

Insert the following in the first paragraph of the result section(line 170-).

“DNA was used for detecting mutations, while RNA was used for detecting fusion genes. This is because when DNA is used for f detecting fusion genes, it is necessary to determine the sequence of the intron region extensively, whereas in the case of RNA, only the sequence of the fusion site is required.”

  1. This compact panel can have global impact beyond Japan. Are the variants tested in this panel occurring at similar frequencies in other countries? Such data can demonstrate how applicable the test is outside Japan.

Our response

We inserted the following in the third paragraph of the discussion section (line 331-).

“EGFR mutations are estimated to be 12.8% in Europeans but 49.1% in Asians; the high prevalence of EGFR mutations is one reason for the emphasis on genetic testing in Japan. Another reason is the difference in specimen collection methods.”

Reviewer 2 Report

The manuscript by Kato and colleagues describes the development and validation of a companion diagnostic test for the sensitive and accurate detection of actionable mutations in NSCLC samples.

Indeed, respect to currently-available NGS-based genes panel anlyses that analyze several genes simultaneously, they focused on the well-established mutations that, if present may drive therapeutic choices and patients prognosis. Thus, the authors designed a focused panel for the study of 5 mutations with a module scheme based on DNA/RNA analysis according on the kind of mutation.

This approach is interesting and the manuscript is well written, however I would like to address some minor points:

- a flowchart summarizing the analytical procedure may be useful

- in the paragraph regarding concordance test results, it is well explained the cause of discordant data between the compact panel and the reference method. The same should be done also for other discrepancies according to table 4

- genes name should be in italics throughout the manuscript.

Author Response

This approach is interesting and the manuscript is well written, however I would like to address some minor points:

- a flowchart summarizing the analytical procedure may be useful

Our response

We made a new Figure 1, and inserted in the end of the section “3.1. Design of the compact panel”.

- in the paragraph regarding concordance test results, it is well explained the cause of discordant data between the compact panel and the reference method. The same should be done also for other discrepancies according to table 4 (line 282-).

Our response

There were no clear reasons for the discrepancy of other samples. We inserted the following sentence a the end of the section

“For the other discordant samples, the cause of the discrepancy was not clear.”

 genes name should be in italics throughout the manuscript.

Our response

We use italic fonts for gene names, and normal fonts for protein names and some product names. Some mistakes were fixed.